# Differential Regulation of Human Serotonin Receptor Type 3A by Chanoclavine and Ergonovine

**DOI:** 10.3390/molecules26051211

**Published:** 2021-02-24

**Authors:** Sanung Eom, Woog Jung, Jaeeun Lee, Hye Duck Yeom, Shinhui Lee, Chaelin Kim, Heui-Dong Park, Junho H. Lee

**Affiliations:** 1Department of Biotechnology, Chonnam National University, Gwangju 61186, Korea; yeomself2355@gmail.com (S.E.); jaeeun3023@gmail.com (J.L.); dltlstn39@gmail.com (S.L.); chaelinkim0215@gmail.com (C.K.); 2School of Food Science and Biotechnology, Kyungpook National University, Daegu 41566, Korea; crazycell79@nate.com; 3GoPath Laboratories, Buffalo Grove, IL 60089, USA; hyeduck@gmail.com

**Keywords:** ergot alkaloids, serotonin receptor, chanoclavine, irritable bowel syndrome

## Abstract

Irritable bowel syndrome (IBS) is a chronic disease that causes abdominal pain and an imbalance of defecation patterns due to gastrointestinal dysfunction. The cause of IBS remains unclear, but intestinal-brain axis problems and neurotransmitters have been suggested as factors. In this study, chanoclavine, which has a ring structure similar to 5-hydroxytryptamine (5-HT), showed an interaction with the 5-HT_3A_ receptor to regulate IBS. Although its derivatives are known to be involved in neurotransmitter receptors, the molecular physiological mechanism of the interaction between chanoclavine and the 5-HT_3A_ receptor is unknown. Electrophysiological experiments were conducted using a two-electrode voltage-clamp analysis to observe the inhibitory effects of chanoclavine on *Xenopus* oocytes in which the h5-HT_3A_ receptor was expressed. The co-application of chanoclavine and 5-HT resulted in concentration-dependent, reversible, voltage-independent, and competitive inhibition. The 5-HT_3A_ response induced by 5-HT was blocked by chanoclavine with half-maximal inhibitory response concentration (IC_50_) values of 107.2 µM. Docking studies suggested that chanoclavine was positioned close F130 and N138 in the 5-HT_3A_ receptor-binding site. The double mutation of F130A and N138A significantly attenuated the interaction of chanoclavine compared to a single mutation or the wild type. These data suggest that F130 and N138 are important sites for ligand binding and activity. Chanoclavine and ergonovine have different effects. Asparagine, the 130th amino acid sequence of the 5-HT_3A_ receptor, and phenylalanine, the 138th, are important in the role of binding chanoclavine, but ergonovine has no interaction with any amino acid sequence of the 5-HT_3A_ receptor. The results of the electrophysiological studies and of in silico simulation showed that chanoclavine has the potential to inhibit the hypergastric stimulation of the gut by inhibiting the stimulation of signal transduction through 5-HT_3A_ receptor stimulation. These findings suggest chanoclavine as a potential antiemetic agent for excessive gut stimulation and offer insight into the mechanisms of 5-HT_3A_ receptor inhibition.

## 1. Introduction

Chanoclavine (Figure 1A) is an ergot alkaloid that is biosynthetically derived from indole compounds produced from l-tryptophan and is a metabolite of fungi of the genus *Claviceps*. Ergots belong to the genus *Claviceps* and are mainly present in grass and grain and can cause infectious diseases when ingested [1]. However, the tetracyclic structure of ergot alkaloids has structural similarities with norepinephrine, dopamine, and serotonin [2], so these alkaloids may act on monoamine receptors and have the potential to treat diseases caused by signaling problems by interacting with neuroreceptors present in synapses when used in the appropriate amount. The general partial structure of ergot alkaloids is a tetracyclic ring system called ergoline, and these alkaloids are divided into clavines, lysergic acid amides, and peptides (ergopeptines) [3]. Ergotamine and dihydroergotamine, belonging to the ergopeptine group, are agonists of serotonin receptors and are used to treat hypotension by stimulating smooth muscles to cause vascular contractions [4,5]. The histamine that acts as central sympathetic neuron activation and peripheral α1-adrenergic blocking prevents migraines [6]. Bromocriptine affects uterine contractions by promoting glutamate uptake in the human glutamate transporter-1 [7] and is related to Parkinson’s disease through potent agonism on the D2 receptor [8,9]. Nicergoline, one of the ergot alkaloids, was shown to be involved in glutamate transport, reducing glutamate levels by accelerating the glutamate uptake [10]. Another alkaloid, metergoline, regulated the voltage-gated ion channel and C-type inactivation of the Kv1.4 channel and demonstrated the potential as a drug through neuronal excitation and the regulation of cardiovascular activation [11]. Previous studies reported that various types of ergot alkaloids had neurotransmitter receptor activity and the potential to be developed as therapeutic agents.

Serotonin (5-HT, 5-hydroxytryptamine) receptors include the seven families of 5-HT_1_, 5-HT_2_, 5-HT_3_, 5-HT_4_, 5-HT_5_, 5-HT_6_, and 5-HT_7_. Of these, 5-HT_3_ is the only ligand-gated ion channel, and all other families are GPCRs [12]. 5-HT promotes or inhibits neurotransmitter (dopamine, cholecystokinin, and GABA) releases by the activation of serotonin receptors in the central and peripheral nervous systems [13]. 5-HT_3_ serotonin receptors, which are ligand-gated ion channel types, regulate ion movement and are involved in electrophysiology characteristics such as selectivity, channel rectification, and conductance. The 5-HT_3_ receptors consist of a cys-loop structure and five subunits surrounding an ion-conducting pore and are classified depending upon their subunits, which may have homomeric (3A) or hetero-pentameric (3B) channels [14]. The main therapeutic application for the 5-HT_3_ receptor is emesis induced in the vomiting center by chemical triggers such as anticancer drugs. The activation of the 5-HT_3_ receptors regulate peristaltic movement and intestinal motility by transporting sensory signals of the sympathetic and parasympathetic systems to the gastrointestinal tract and the intestinal nervous system. [15]. In the central nervous system, 5-HT_3_ receptors are involved in emesis, recognition, anxiety, and depression. In particular, these receptors are present at high levels in the brain stem and expressed in many other brain regions such as the postrema hippocampus, nucleus accumbens, ventral tegmental area, and nucleus tractus solitarius. Among them, the hippocampus, nucleus accumbens, and ventral tegmental area relate to the vomiting reflex [16,17,18]. Since serotonin mediates various physiological effects, psychiatric disorders or serotonin syndrome can occur from insufficient or excessive serotonin in the human body [19]. Therefore, specific serotonin receptor subtypes that mediate serotonin signaling can be used as therapeutic targets to regulate serotonin interactions.

Previous research has analyzed the structural features of ergot alkaloids related to the interaction between receptors and ergot alkaloids. It is known that hallucinogens such as lysergic acid diethylamide (LSD) interact with 5-HT_1A_ and 5-HT_1B_, as well as 5-HT_2A_, 5-HT_2B_, and 5-HT_2C_, in the brain [20]. Ergopeptines and clavines interact with different 5-HT-receptor subtypes. The ergopeptine group showed strong activity with 5-HT_1A_, 5-HT_1B_, 5-HT_1D_, 5-HT_2A_, and 5-HT_2C_, and the clavine group showed strong activity at rat 5-HT_2A_, human 5-HT_1D_, and 5-HT_1F_ [21].

Chanoclavine is a precursor of argroclavine, elymoclavine, and lysergic amide, which are tetracyclic ergolines. Although the pharmacological effects of chanoclavine have not yet been precisely determined, its derivative (KSU-1415; 4,5-trans-5-(2-methyl-1-propen-1-yl)-4-propylamino-1,3,4,5-tetrahydrobenz(cd)indole) binds to dopamine receptors and affects neurological disorders by regulating dopamine signaling [22]. Elymoclavine was also reported to be related to dopaminergic agonist action [23]. Studies of the chanoclavine derivatives suggested them as possible ligands from structural similarity to the chanoclavine precursor [21,22].

This study showed that, among the ergot alkaloids, chanoclavine might demonstrate a therapeutic potential by regulating the 5-HT_3A_ receptor activity. Although previous studies on the derivatives of chanoclavine and their analogs showed pharmacological effects by stimulating the monoamine receptor, no effects on the 5-HT_3_ receptor were reported. We confirmed the serotonin activation of the inward current using a two-electrode voltage clamp after injecting human 5-HT_3A_ receptor mRNA into *Xenopus* oocytes [24]. When treated with chanoclavine and serotonin together, the inward peak current was considerably inhibited. We investigated whether chanoclavine and ergonovine bound competitively to the 5-HT_3_-receptor binding site and determined the location of the binding site using point mutations. Finally, we compared in silico the binding mode of chanoclavine and granisetron (a 5-HT_3A_-receptor antagonist). 

## 2. Results

### 2.1. Effects of the Ergot Alkaloid on Human 5-HT_3A_ Receptor Activation Current

To examine the effects of chanoclavine and ergonovine, a 5-HT current was elicited using *Xenopus* oocytes injected with 5-HT_3A_ mRNA. 5-HT (100 µM) in bath solution flowed into human 5-HT_3A_ receptors in *Xenopus* oocytes at 2 mL per minute, and then, the 5-HT currents were inwardly rectified through a two-electrode voltage clamp. When treated with only chanoclavine or ergonovine, there was no change of the −80 mV holding potential (data not shown). However, the treatment with chanoclavine and serotonin reduced the inward current peak of serotonin, indicating that 5-HT_3A_ receptors were partially blocked (Figure 1B). Unlike chanoclavine, ergonovine did not produce changes, showing the same value as the 5-HT-stimulated inward current (I_5-HT_). Thus, chanoclavine may specifically interact with certain amino acid residues in the 5-HT_3A_ receptors compared to other alkaloids. When treated with ergonovine and chanoclavine at the same concentration (100 µM) as serotonin, the inhibition of ergonovine and chanoclavine was 1.1 ± 4.8% and 42.5 ± 9.7%, respectively (Figure 1C).

### 2.2. Chanoclavine Induces the Concentration-Dependent Inhibition of I_5-HT_

Oocytes expressing human 5-HT_3A_ receptors were exposed to various concentrations of ergonovine (EG) or chanoclavine (EC) and 100 µM 5-HT. The co-application of ergonovine with 5-HT did not show any noticeable reduction and no change in the current as the concentration increased (Figure 2A,B). In contrast, chanoclavine induced a lower current than the treatment with 100 µM 5-HT alone and showed a reduced rate of about half at the same ratio of 5-HT and EC (Figure 2C,D; half-maximal inhibitory response concentrations (IC_50_) = 107.2 ± 22.8). Figure 2C,D shows the dose-dependent effect of chanoclavine. The maximum inhibition (I_max_) was 83.6 ± 12.6%, and the inhibition of I_5-HT_ was 4 ± 3.8%, 7.9 ± 4.1%, 42.5 ± 5.7%, 61.6 ± 6.1%, and 75 ± 4.5% at 10, 30, 100, 200, and 300 µM, respectively, in oocytes expressing the 5-HT_3A_ receptor (*n* = 8–10 from four different frogs).

### 2.3. Chanoclavine Inhibits Attenuation of the I_5-HT_ in 5-HT_3A_ Mutant

This study showed that I_5-HT_ was competitively inhibited by chanoclavine in a concentration-dependent manner. This result indicates that chanoclavine interacted with the same binding site of 5-HT. To identify the particular amino acid positions regulating the I_5-HT_ inhibition, amino acids were modified by site-directed mutations. Single-point mutations to alanine were introduced, and the mutated h5-HT_3A_ receptor was expressed in oocytes (Figure 3A–D). While Y91A showed similar currents as the wild-type 5-HT_3A_ current (Figure 3A), the inhibition currents of F130A and N138A were attenuated at 100 µM 5-HT and chanoclavine (Figure 3B,C). Based on the attenuating effect of F130A and N138A, a double mutation was performed using F130A as a template to identify the specific binding of single F130A or N138A alone mutants. At 300 µM chanoclavine with 5-HT, the inward current was not significantly decreased, similar to the 5-HT-only-induced current (Figure 3D), and the inhibition curves of the wild type and mutants were fitted according to Hill’s equation. Additional mutants of the human 5-HT_3A_ receptors are shown in Table 1.

### 2.4. Current-Voltage Correlation of 5-HT_3A_ Receptors and Competitive Inhibition of I_5-HT_ Current by Chanoclavine in Wild Type and Mutants

In this experiment to study the relationship between the current and voltage, the membrane potential was held at −80 mV, and traces were obtained by ramping from −80 to +60 mV over a duration of one sweep per two seconds using the voltage ramp protocol. Before the 5-HT treatment (Figure 4A, control), the current from the exposure to the bathing solution alone was insignificant in negative and positive voltages (both were near −0.1 and +0.1 µA). In oocytes flowing with the bath solution and 5-HT, 5-HT evoked more inward and outward currents than the control, and their reversal potential was close to 0 mV. Figure 4A−D represents the currents of the wild-type h5-HT_3A_ and the F130A, N138A, and F130A+N138A mutants. The currents of the mutants treated with 100 µM 5-HT and EC were similar to the treatment with 100 µM 5-HT alone, whereas the current of the control group decreased. The wild-type 5-HT_3A_ voltage ramp current activated by 5-HT and EC was inhibited 42.5 ± 2.3%, 40.9 ± 1.1%, 47.9 ± 3.4%, and 49.5 ± 4% at −60, −30, +30, and +60 mV holding potential, showing that the 5-HT_3A_ inward current was independent upon the holding potential voltage.

To explore the role of chanoclavine in the mechanism, the inhibitory effect was analyzed by the treatment with various concentrations of 5-HT while the concentration of chanoclavine was fixed (Figure 4E). This experiment was performed to determine the effect of 100 and 30 µM chanoclavine on the 5-HT peak current induced by different concentrations of 5-HT in *Xenopus* oocytes expressing 5-HT_3A_ receptors. The curves were normalized with the I_max_ of the 5-HT-induced current, and the control current according to each concentration was used as the standard. When competing at the same rate, chanoclavine showed a current of about 50% of the control group. The presence of different 5-HT concentrations had a significant effect on the dose-response curve of 5-HT in oocytes expressing 5-HT_3A_ receptors. Figure 4E indicates that chanoclavine interacted with the 5-HT-binding site and regulated the 5-HT_3A_-receptor channel activity in a competitive manner (*n* = 8–10 from four different frogs).

### 2.5. Complex Docking Modeling of the 5-HT_3A_ Receptor with Chanoclavine

The covalent docking protocol was used to predict the amino acid residues involved in the interaction of chanoclavine with the 5-HT_3A_ receptor. A docking simulation was used to analyze the potential binding locations and perform the modeling required for the accurate binding energy calculations of chanoclavine with the 5-HT_3A_ receptor. The homology models with mutants F130A and N138A were built on the template of the wild-type 5-HT_3A_ receptor. Only those residues whose interaction function was confirmed to be lost through point mutation and a two-electrode voltage clamp were considered for atomic distances using Pymol (Figure 5G,H). The template known as the human 5-HT_3A_ receptor used in BLAST showed 98.6% identity and was selected as a suitable template (ID code 6Y1Z, 2.82-Å resolution). The molecules were modeled through movements such as twisting, angle, and rotation at the binding site. The modeling structure of chanoclavine was simulated in the 5-HT_3A_ receptor to show the best-fitting position (Figure 5). Figure 5 shows that chanoclavine is docked in the binding pocket area on the extracellular side of the 5-HT_3A_ receptor (Figure 5A,B). The docking results of Figure 5C,D are: Binding Energy = −4.59 kcal/mol, Intermolecular Energy = −5.79 kcal/mol, Torsional Energy = 1.19 kcal/mol, Unbound Extended Energy = −0.54 kcal/mol, and refRMS = 231.78). The inverse logarithm of inhibition constant was pKi = 3.37. The most ideal docking conformation indicated that chanoclavine could form strong hydrogen bonds with the wild-type 5-HT_3A_ receptor. However, in the mutants, each residue replaced with alanine had a weaker binding than the wild type. F130, N138, and L156 residues were used as interacting site residues (F130 distance = 3.4, 3.8, 3.4, and 3.6 Å; N138 distance = 3.1, 4.2, and 4.3 Å; and L156 distance = 3.7, 3.8 and 4.1 Å in the wild type and F130A distance = 7.0, 5.8, and 6.3 Å; N138A distance = 3.8 and 4.5 Å; and L156A distance = 5.2 and 6.1 Å in the mutant type). The mutant type appears to increase considerably the interatomic distance and decrease the number of possible bonds compared to the wild type. However, the currents of all the other 5-HT_3A_ mutants showed no effect from chanoclavine (Figure 3 and Table 1). Y153 plays a critical role in the binding and gating of serotonin and antagonists (granisetron, etc.) at the binding pocket of 5-HT3A, because tyrosine can make the hydrogen bond with the ligand [25]. When comparing our results with the binding sites of existing papers [26,27], chanoclavine is a competitive inhibitor that acts on the same binding pocket. However, F130 and N138 are the key residues for binding, not Y153 (Table 1 and Figure 3). In conclusion, chanoclavine binds to the same binding pocket as the existing agonists and antagonists, but chanoclavine interacts with the novel residues F130 and N138, which are different from those of the well-known residues (I71, Y91, Y153, I228, etc.) [26,27].

## 3. Discussion

The issues of ergot alkaloid toxicity are well-known, but at the same time, ergot has long been used as a medicine [28]. The reason can be found in the strong agonism and antagonism of ergot alkaloids on dopamine, serotonin, and adrenaline receptors [29]. Therefore, severe symptoms can be treated using an appropriate dose. However, their use should be managed legally, such as by a doctor’s prescription, to prevent abuse and misuse and ensure proper use according to the patient’s condition. Many drugs in the ergoline alkaloid family have been approved by the United States Food and Drug Administration (FDA) [30]. Most of these drugs interact with the 5-HT_1_ and 5-HT_2_ subtypes in the 5-HT family. Representatively, dihydroergotamine and ergotamine act on the 5-HT_1_ and 5-HT_2_ receptors to constrict blood vessels and suppress nerve inflammation in the dura mater. Thus, they are used only by a doctor’s prescription to treat severe migraine headaches (not a simple headache) or in emergencies.

In particular, ergonovine was reported to show a high affinity for 5-HT_1A_, 5-HT_1B_, 5-HT_1D_, 5-HT_2A_, and 5-HT_2C_. On the 5-HT_2_ receptor, it acted as a partial agonist at lower concentrations (0.1–1 µM) and a competitive antagonist at 10 µM [31]. It is a medicine used to treat severe bleeding after childbirth. However, chanoclavine has only been identified to modulate dopamine D2 receptors in the brain, and there is no proven interaction with the 5-HT family. Of the clavine group to which chanoclavine belongs, only lysergol interacts with 5-HT_2A_. In other words, chanoclavine has no clinical trial, and studies of the mechanisms of the neuronal receptors at the molecular cell level are lacking.

The clavine group showed an interaction with rat 5-HT_2A_ receptors, human 5-HT_1D_ receptors, and 5-HT_1F_ receptors, but no studies have reported that chanoclavine interacts with other members of the 5-HT family. Considering the above, we wondered if chanoclavine might interact with the 5-HT_3_ receptor, and, if not, the interactions with other 5-HT family members are essential in determining its proper usage. In addition, in an in vivo toxicity study using mice, no toxicologically significant effects on gross hematology, blood chemistry, pathology, and histology were seen and showed chanoclavine to be of low toxicity and raised no food safety concerns [32]. These results indicate that chanoclavine has the potential for development as a therapeutic agent.

Serotonin is a monoamine neurotransmitter that modulates cognition functions and many other physiological functions, such as vomiting and vasoconstriction. Its signals are transmitted through various receptors, affecting specific targets mediated by each receptor [33]. In the human body, most serotonin exists in enterochromaffin cells in the gastrointestinal tract, which are involved in the regulation of gastrointestinal motility. 5-HT_3A_, one of the serotonin receptors, is the main therapeutic target in IBS and emesis, caused by chemical triggers such as anticancer drugs [34]. The 5-HT_3A_ receptor transmits the signal of serotonin, released from enterochromaffin cells to the central neuron system. The antagonism on the 5-HT_3A_ receptor can reduce the over-signaling of serotonin in the gastrointestinal tract. These points can mediate the therapeutic effect. Therefore, we used the potential of ergot alkaloids as a neurotransmitter substitute to examine the activity of the 5-HT_3A_ receptor with different ligands.

Through an in silico docking simulation, we identified possible binding site candidates. Thus, in order to demonstrate the real interaction of chanoclavine with the 5-HT_3A_ receptor, point mutation and electrophysiology experiments were performed. In this study, we evaluated if ergot alkaloids would interact with 5-HT receptors, because alkaloids have a structure that might resemble 5-HT. We screened a candidate group of ergot alkaloids and observed that chanoclavine had a regulatory effect on the 5-HT_3A_ receptor. Chanoclavine, with a tricyclic ring structure, was studied to investigate how it interacted with the 5-HT_3A_ receptor. Human 5-HT_3A_ receptor mRNA was injected into *Xenopus* oocytes, and the regulation of the channel activity in the oocytes was evaluated.

These experiments produced several findings. (1) Chanoclavine reduced the 5-HT-induced peak current in the concentration-dependent and reversible manners (Figure 2). Inhibition of the 5-HT-induced current peak by chanoclavine was reversible, and the peak value induced by 5-HT with chanoclavine decreased as the chanoclavine concentration increased. These results indicate that chanoclavine could regulate the ion efflux of homomeric 5-HT_3A_ receptor channel activity. Additional experiments were performed to determine how chanoclavine suppressed the activity of the 5-HT_3A_ receptor, which is a ligand-gated ion channel. (2) Chanoclavine inhibited the 5-HT-induced current in voltage-independent and competitive manners (Figure 4). When the voltage ramp changed from −80 to +60 mV, the reverse potentials were close to 0 mV, and the trace was not affected by the voltage, because it did not change abruptly at a certain potential. As shown in Figure 4E, chanoclavine acted as a competitive modulator by blocking the 5-HT-binding sites of the 5-HT_3A_ receptors. However, it did not completely block the binding sites, because it did not suppress all the channel ion flow activated by 5-HT (Figure 2). In the competitive experiment, 5-HT at high concentrations exceeded the effects of chanoclavine, indicating that chanoclavine and 5-HT shared the same binding site. (3) The amino acids F130 and N138 of the 5-HT_3A_ receptor were very likely involved in the interaction with chanoclavine (Figure 3 and Figure 5). Based on the inhibition of the 5-HT channel activity by a structure similar to chanoclavine, a mutation was performed by selecting a position supposed to be a binding site. In the in-silico study, the best fit in the binding pocket was shown through molecular docking, and chanoclavine predominantly interacted with specific residues of the 5-HT_3A_ receptor (Figure 5). To confirm the regulatory effect of chanoclavine for each residue, the analysis showed that other mutants except for F130 and N138 maintained their activation. Unlike mutations that were similar or more inhibited than the wild type, when the amino acids at the F130 and N138 positions were changed to alanine, the inhibitory function was lost (Figure 3). Additionally, the double mutation of N138A and F130A showed a further offset of the binding capacity of chanoclavine. The site-directed point mutagenesis method allowed us to identify the ligand-binding sites. We made the point-mutant 5-HT3A receptors by transforming all the amino acids of the candidate residue to alanine. Among them, two mutant types (F130A and N138A) only lost the function of chanoclavine while the function of serotonin remained. As mentioned, chanoclavine was a competitive inhibitor, and chanoclavine residues bound to the F130 and N138 ligand-binding sites located close to loops A and E in the extracellular domain located in the serotonin-binding site [35].

Previous studies showed that molecules within the alkaloid structures were associated with competitive antagonists. Methadone and quinine, used to treat drug addiction and as analgesics, are representative examples of voltage-independent and competitive 5-HT_3A_-receptor antagonists. Since they have competitive properties, 5-HT_3_ antagonists that bind to the orthosteric ligand-binding site were used to define the mechanism by allowing docking to the binding site model. As there was no effect on the voltage, the results suggest that the binding sites did not block the ion pores and shared the same binding site through a structural similarity with 5-HT.

Ergonovine, one of the lysergic acid amides, is a 9-ergolene type of ergot alkaloid. Although ergonovine and chanoclavine are ergot alkaloids with a similar ergoline structure, these two alkaloids had opposing effects (Figure 1 and Figure 2). Except for the common structure ergoline, ergonovine has CONHCH(CH_3_)CH_2_OH and CH_3_ and chanoclavine has CH_2_OH and H in its R1 and R2 residues. The results and structural differences indicate that chanoclavine was specifically docked to the binding site, unlike other alkaloids.

In summary, chanoclavine inhibited the 5-HT-induced current in concentration-dependent, reversible, voltage-independent, and competitive manner. Despite the structural similarities between EG and EC, different effects were observed, and the amino acid residues of EC were considered to be important for the interaction with the 5-HT_3A_ receptor. Based on the site-directed mutagenesis, F130 and N138 of the 5-HT_3A_ receptor could be important positions for the regulation of 5-HT_3A_ activity and function. These 5-HT_3A_ receptor antagonists suppressed not only excessive peristalsis but, also, the induction of vomiting by the brain suppressing serotonin released from the gastrointestinal tract. These inhibitory effects of chanoclavine showed potential as pharmacological agents by regulating the signals in the nervous system.

## 4. Materials and Methods

### 4.1. Materials

Human 5-hydroxytryptamine (serotonin) receptor 3A cDNA (Genebank number: BC004453) was obtained from OriGene (Rockville, MD, USA). Figure 1A represents the chemical structure of two ergot alkaloids, EG and EC, obtained from Sigma-Aldrich (St. Louis, MO, USA). These ergot alkaloids were diluted in bath solution after dissolving in dimethyl sulfoxide (DMSO) for use as stock solutions. The final treatment concentration of DMSO was less than 0.01%. *Xenopus laevis* frogs were obtained from the Korean Xenopus Resource Center for Research (KXRCR000001, Chuncheon, Gangwondo, Korea), and other agents were purchased from Sigma-Aldrich.

### 4.2. In Vitro Transcription and Site-Directed Mutagenesis of the Human 5-HT_3A_ Receptor

The site-directed mutation of one or two amino acids was performed using sense and antisense primers and Pfu polymerase (QuickChange Site-Directed Mutagenesis Kit; Agilent, Santa Clara, CA, USA) and amplified by polymerase chain reaction (PCR) to increase the target domain. After removing the existing methylated cDNA using Dpn Ⅰ, the PCR products were transformed into *Escherichia coli* strain DH5α. The cDNA of the 5-HT_3A_ receptor was linearized by Xho Ⅰ at the end of the multi-cloning site, followed by transcription using T7 polymerase and a transcription kit (mMESSAGE mMACHINE T7 Transcription Kit; Thermo Fisher Scientific, Waltham, MA, USA). The RNA product was dissolved in RNase-free water and stored at −80 °C.

### 4.3. Isolation of Xenopus Oocytes and Human 5-HT_3A_ mRNA Microinjection

To isolate oocytes, female *Xenopus laevis* were put into a container filled with ice and left for about one hour. When the blood vessels contracted and fell into a state similar to hibernation, the abdomen was incised, and the ovaries were removed and cut into small pieces in Ca^2+^-free OR2 solution (82.5 mM NaCl, 2.5 mM KCl, 1 mM MgCl_2_, and 5 mM HEPES, pH 7.4) with 2 mg/mL collagenase. After a 2 h agitation to digest the follicle membranes, isolated oocytes in stages Ⅴ and Ⅵ were selected and washed repeatedly. The selected oocytes were maintained in ND96 incubation solution (96 mM NaCl, 2 mM KCl, 1.8 mM CaCl_2_, 1 mM MgCl_2_, 5 mM HEPES, 2.5 mM sodium pyruvate, and 50 µg/mL gentamycin, pH 7.4) at 16–18 °C. The solution containing the oocytes was changed twice a day and stored in a shaking incubator. Under the microscope, the end of a glass capillary was broken and fitted with a 20 µm diameter needle and filled with mineral oil, discharging air bubbles from the tip. When the *Xenopus* oocytes were ready for injection, 50 ng of mRNA was injected with a nanoliter injector (Drummond Scientific, Vernon Hills, IL, USA). The recording was started after two days of sufficient shaking incubation in ND96 solution. Surgery, microinjections, and the other handling of *Xenopus laevis* followed the standard protocols [36,37,38].

### 4.4. Voltage Clamp Data Recording

The oocytes in which mRNA was expressed during the incubation period allowed for the estimation of channel or receptor properties for observing drug and ligand potential interactions using a two-electrode voltage clamp (OC-725C; Warner Instruments, Hamden, CT, USA) and Digidata (1322A; Molecular Devices, Sunnyvale, CA, USA). The two-electrode voltage clamp had voltage and current electrodes that stably maintained the potential and delivered the transmembrane potential and high current. The voltage clamp amplifier transmitted the high current of the command potential, followed by the Digidata conversion of the analog signal of the amplifier to one capable of reading. A computer transferred the converted output amplitude of the voltage clamp amplifiers to an input digital converter using pClamp 10 software (Axon Instruments, Union City, CA, USA). The voltage and current electrodes were filled with 3 M KCl (0.3–0.7 MΩ) and analyzed at a −80 mV holding potential. The oocytes were placed in the chamber and exposed to a ND96 solution at a flow rate of 2 mL per minute. In the electrophysiological experiments, the voltage ramp recording was performed at room temperature, and a current from −80 to +60 mV was applied to determine the current and voltage relationships. Inward peak traces of the 5-HT_3A_ receptor and voltage ramp traces were converted to a value through Clampfit 9.0 (Molecular Devices, San Jose, CA, USA).

### 4.5. Modeling and Molecular Docking

For the molecular docking studies of the model, the 5-HT_3A_ receptor from BLAST with 98.6% identity was used. The adapted template and protein structure of the 5-HT_3A_ receptor were obtained from the Protein Data Bank (ID code 6Y1Z, 2.82-Å resolution). The three-dimensional structure of the ligand (chanoclavine) was referenced in PubChem (ID code 5281381). Molecular docking studies were programmed considering the intermolecular energy, the inhibition constant, crystal structures, and energy minimized using Autodock Tools of The Scripps Research Institute (version 4.2.6, La Jolla, CA, USA). Chanoclavine was docked with the basic settings of the Autodock program, except for a few settings. We removed the water from the macromolecule, added polarity and hydrogen, and charged the compute gasteiger. The grid box information was as follows: center (x, y, z); (128.044, 128.03, 162.037), number of points (x, y, z); (49, 51, 59), and spacing, 1.0. The protein complex of chanoclavine and the 5-HT_3A_ receptor was analyzed using Ligplot (version 4.5.3, EMBL-EBI, Hinxton, Cambridgeshire, UK) and Pymol (version 1.8.4.2, Schrödinger, New York, NY, USA). Ligplot showed a binding activity between the ligand and protein. Pymol was used to measure the space between chanoclavine and the mutant amino acids of the 5-HT_3A_ receptor.

### 4.6. Data Analysis

In the experiments, all data were represented as the mean ± SEM (standard error of the mean) of the 5-HT-stimulated inward peak in the 5-HT_3A_ receptors. Concentration-dependent curves of chanoclavine and ergonovine were fitted according to the Hill equation: y = V_min_ + (V_max_ − V_min_) × [X]^n^/(IC_50_^n^ + [X]^n^), where y is the serotonin-induced peak amplitude at various chanoclavine concentrations, V_min_ and V_max_ are the minimum and maximum values, respectively, [X] is the chanoclavine or 5-HT concentration, IC_50_ is the half-maximal inhibitory response concentration of chanoclavine, and n is the interaction coefficient. All curves plotted through the Hill equation were generated using Origin Pro 7.0 (OriginLab, Northampton, MA, USA). The significance of differences between the control and chanoclavine treatment was determined according to the *p*-value in the Student’s *t*-test. *p*-values of less than 0.05 were considered statistically significant.

## Figures and Tables

**Figure 1 molecules-26-01211-f001:**
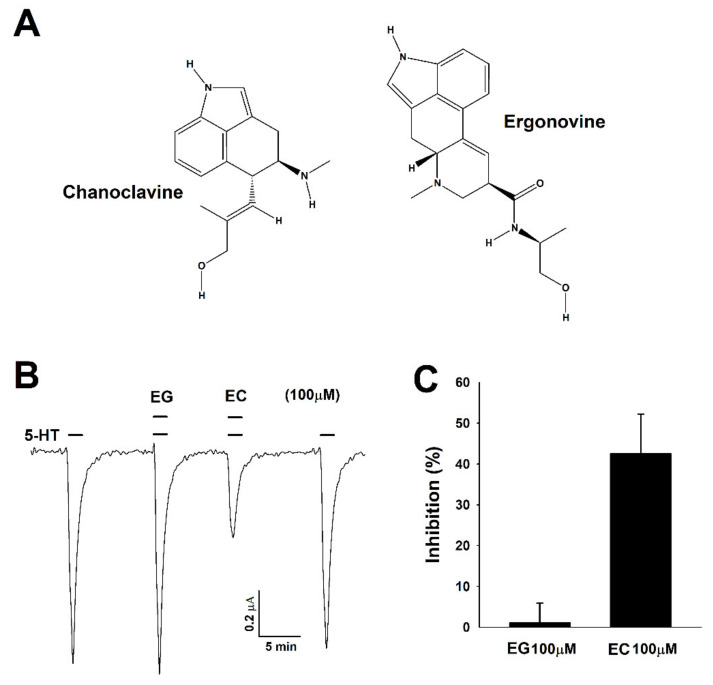
Chemical structure of chanoclavine and ergonovine and the inhibitory effects on *Xenopus* oocytes expressing human 5-hydroxytryptamine (5-HT)_3A_ receptors. (**A**). Structure of chanoclavine (EC) and ergonovine (EG). (**B**). After flowing 5-HT mixed with bath solution for one minute per 2 mL, 5-HT (100 µM) induced a reversible inward current (I_5-HT_). The traces indicate the inward current in the presence of 5-HT coapplied with ergonovine and chanoclavine at 100 µM. The representative traces from four different frogs elicited at the −80-mV holding potential. (**C**). The histogram indicates that the percentage of inhibition by chanoclavine was 42.5 ± 9.7 calculated from the mean of the peak inward current. Each value represents the mean ± S.E.M (*n* = 8–10 from four different frogs).

**Figure 2 molecules-26-01211-f002:**
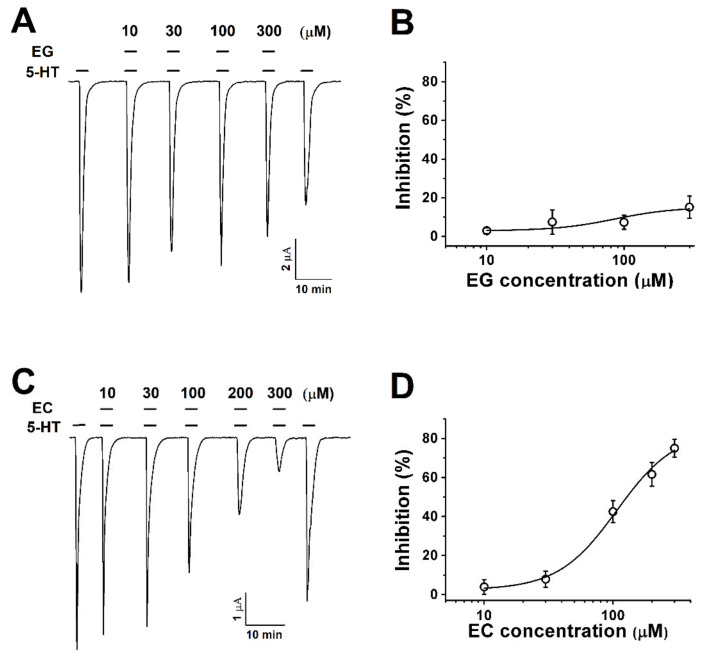
The concentration-dependent response of *Xenopus* oocytes expressing human 5-HT_3A_ receptors to ergot alkaloids. (**A**). The traces represent increases in the current inhibition with increases in the ergonovine concentration. Treatment with 5-HT (100 µM) alone induced an inward current followed by various response traces with 5-HT (100 µM) and ergonovine (10, 30, 100, and 300 µM). (**B**). The plot shows the ergonovine inhibition current fitted according to Hill’s equation, where the maximum inhibition (I_max_) was 15 ± 5.7%. (**C**). The traces induced by chanoclavine with 5-HT (100 µM) appear dependent upon the concentration (10, 30, 100, and 300 µM). (**D**). Average percentage curves of chanoclavine fitted according to Hill’s equation based on the peak of the inward current. The I_max_ was 83.6 ± 12.6%, and the inhibition was 4 ± 3.8%, 7.9 ± 4.1%, 42.5 ± 5.7%, 61.6 ± 6.1%, and 75 ± 4.5% at 10, 30, 100, 200, and 300 µM, respectively. The oocytes expressing human 5-HT_3A_ mRNA were held at −80 mV, and each value represents the mean ± SEM (*n* = 8–10 from four different frogs).

**Figure 3 molecules-26-01211-f003:**
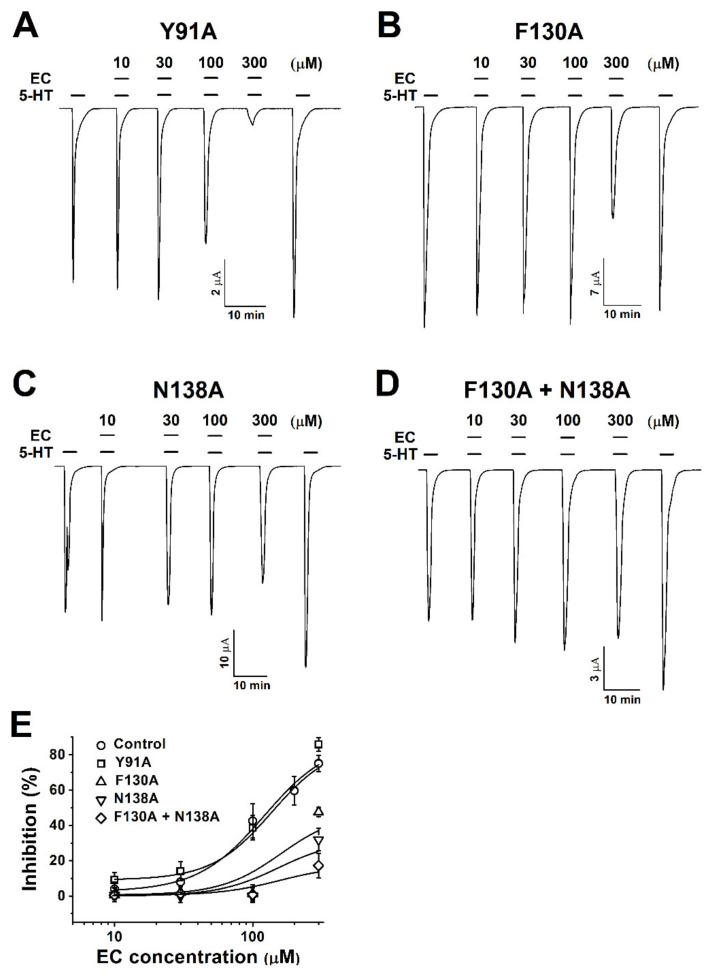
Concentration-dependent inhibition responses of mutants to chanoclavine. (**A**–**D**). Y91A, F130A, N138A, and F130A+N138A mutant receptors are rectified at a −80 mV holding potential. The 5-HT_3A_ mutant receptors expressed in *Xenopus* oocytes elicit reversible currents using an electrode voltage clamp. All oocytes were exposed to 10, 30, 100, and 300 µM chanoclavine coapplied with 100 µM 5-HT after treatment with 5-HT alone. (**E**). The inhibition percentage curve according to the EC concentration of the 5-HT_3A_ mutant receptors. Each value represents the mean ± SEM. Additional half-maximal inhibitory response concentrations (IC_50_), I_max_, and Hill’s coefficient values of the other mutants are listed in Table 1 (*n* = 6–8 from four different frogs).

**Figure 4 molecules-26-01211-f004:**
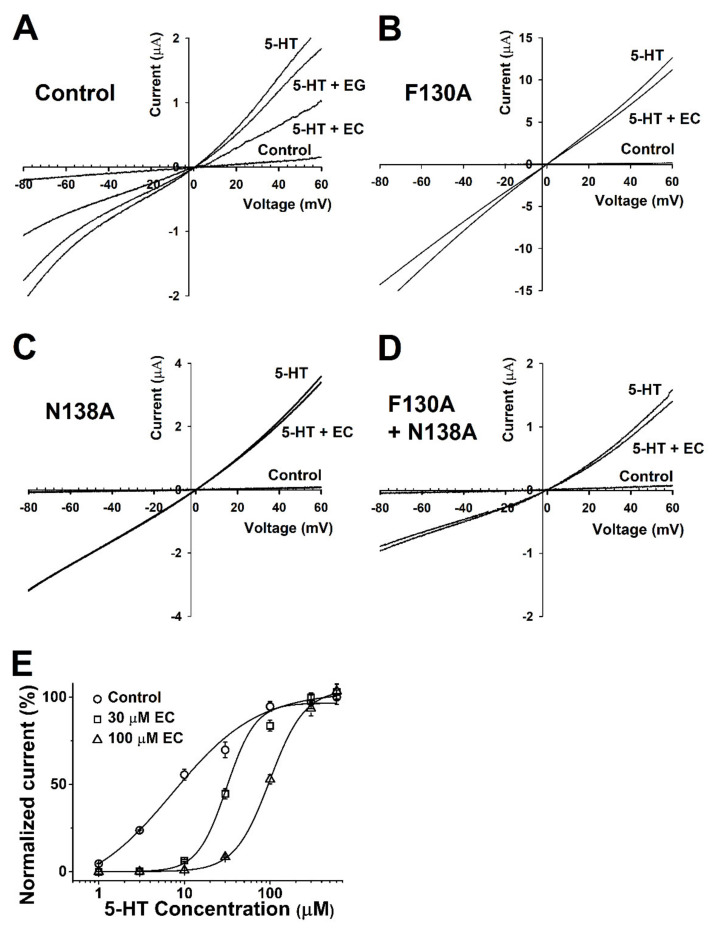
Voltage dependency and competition inhibition of chanoclavine in I_5-HT_ in oocytes expressing 5-HT_3A_ receptors. (**A**–**D**). Current–voltage relationship of I_5-HT_ inhibition by chanoclavine in 5-HT_3A_ receptors. Wild-type 5-HT_3A_ receptor traces showed a current–voltage relationship with EC and EG at the same concentration as 5-HT (100 µM). All control trace voltage ramps change with only the bath solution flowing without treatment. Traces were obtained using voltage changes from −80 mV to +60 mV over a duration of one sweep per two seconds. **B**–**D** Indicate F130A, N138A, and F130A+N138A, respectively. (**E**). Concentration-response effect of 5-HT in the absence or presence of chanoclavine on oocytes expressing human 5-HT_3A_ receptors. Control (○) represents various 5-HT concentration currents from 1 to 100 µM in the absence of chanoclavine, and 5-HT at 30 µM (□) and 100 µM (△) chanoclavine represents the change in the current compared to the control. Normalized currents were divided by the I_max_ of the control current and the I_max_ of 5-HT alone, with 30 µM and 100 µM EC at 99.1 ± 14.2, 94.6 ± 24.4, and 60 ± 2.2, respectively. Oocytes were exposed to 5-HT alone or with EC over a duration of 2 mL per minute and held at a −80 mV membrane holding potential. Each value represents the mean ± SEM (*n* = 8–10 from four different frogs).

**Figure 5 molecules-26-01211-f005:**
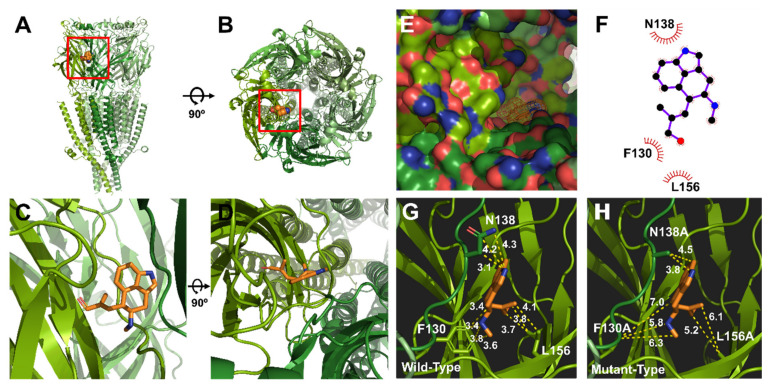
Molecular docking modeling of chanoclavine to the 5-HT_3A_ receptor. (**A**–**D**). Side and top views of the chanoclavine docking model to the 5-HT_3A_ receptor. (**E**). Binding pocket in the 5-HT_3A_ receptor region of the extracellular domain membrane of the pocket side. Chanoclavine docked to the extracellular region that forms a loop complex in the 5-HT_3A_ receptor. (**F**). 2D schematic of the predicted binding mode of chanoclavine in the ligand-binding pocket. (**G**,**H**). Binding interactions of the ligand and residues in the wild and mutant types. The replaced mutants changed the interaction activity to varying degrees.

**Table 1 molecules-26-01211-t001:** Inhibitory effect of 100 µM chanoclavine on wild-type and mutant receptors.

5-HT_3A_ Residue	Inhibition Percentage (%)
Wild type	42.49 ± 5.7
I71A	62.67 ± 3.1
Y91A	50.20 ± 5.1
R92A	95.45 ± 0.3
F130A	9.36 ± 4.3
V133A	67.19 ± 9.9
S136A	68.39 ± 6.1
N138A	−1.64 ± 6.2
I139A	92.83 ± 4.9
Y153A	56.93 ± 3.6
L156A	46.11 ± 6.5
R196A	82.07 ± 2.5
I228A	66.16 ± 9.2

Inhibition percentages represent the mean ± SEM (*n* = 6–8/group). 5-HT: 5-hydroxytryptamine.

## Data Availability

The data presented in this study are available in the article.

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
