# Peer review of "Differential Regulation of Human Serotonin Receptor Type 3A by Chanoclavine and Ergonovine"

_molecules, 2021, doi:10.3390/molecules26051211_

Round 1

Reviewer 1 Report

I made a careful review of the manuscript entitled "Differential regulaiton of ergot alkaloids on humana serotonin receptor type 3A". The work is very interesting but it was equally interesting to clarify and discuss some aspects before the paper can be accepted for publication. The first aspect concerns, as reported in the literature, the possibility that these ergot alkaloids also interact with the 5-HT2 receptors, therefore, given that no mention is made of it, it must be highlighted and understood if this can determine interference on the described potential. therapeutic. Furthermore, a not secondary aspect is the toxicity of these alkaloids and once again the possibility that they will affect the therapeutic potential unless you just want to identify them as new lead compounds. Therefore the study, although interesting, needs to be integrated either with an adequate discussion or with some invivo evidence Therefore on the basis of these considerations the work cannot be accepted in the present form but after the appropriate modifications indicated

Author Response

1# REVIEWER’S COMMENTS

# I made a careful review of the manuscript entitled "Differential regulation of ergot alkaloids on human a serotonin receptor type 3A". The work is very interesting but it was equally interesting to clarify and discuss some aspects before the paper can be accepted for publication.

# The first aspect concerns, as reported in the literature, the possibility that these ergot alkaloids also interact with the 5-HT2 receptors, therefore, given that no mention is made of it, it must be highlighted and understood if this can determine interference on the described potential. therapeutic.

Authors’ response:

Thanks to your proper perspective. We fully agree with your aspects. So we can show thorough consideration of Chanoclavine's potential as a medicine. And we can revise the manuscript in a better way.

We added a reference to the literature on Chanoclavine and Ergonovine interacting with the 5-HT2 group. In addition, the possibility that Chanoclavine is specific to 5-HT3 and why the study in 5-HT3 is important was added to the discussion. – We added two paragraphs with discussion about the two aspects you presented. [page 10, line 282-299]

# Furthermore, a not secondary aspect is the toxicity of these alkaloids and once again the possibility that they will affect the therapeutic potential unless you just want to identify them as new lead compounds. Therefore the study, although interesting, needs to be integrated either with an adequate discussion or with some in vivo evidence Therefore on the basis of these considerations the work cannot be accepted in the present form but after the appropriate modifications indicated

Authors’ response:

The duality of toxicity and drug properties of ergot alkaloids is that the dose is important in the end, and if the excellent efficacy of ergot alkaloids is used in an appropriate way such as prescribed by a doctor, it is an effective drug to treat strong symptoms of difficult to treat with general drugs. Also, many medicines of ergot alkaloids approved by the FDA.

The above contents were added to the discussion. - The first paragraph was corrected with discussion about the two aspects you presented. [page 10, line 270-281]

We hope this revised manuscript will meet your expectations. 

All revised parts are highlighted in red. 

Reviewer 2 Report

GENERAL COMMENTS:

The work presented in this manuscript used the xenopus oocyte model as a tool to study the interaction of chanoclavine with the 5HT3A receptor. The work described appears to be done correctly with adequate replication. My main issue that must be addressed prior to acceptance is that the English needs to be improved throughout manuscript.

Out of all of the ergot alkaloids to choose from to compare with chanoclavine, why was ergonovine selected? Why not another clavine alkaloid or an ergopeptine?

SPECIFIC COMMENTS:

L124 – Should the first figure referenced in the text be Fig. 1A? Perhaps reference Fig. 1A in the introduction where structural aspects are discussed such as L38 or L42.

L314 – This statement made me wonder, what was different in the structure of ergonovine that caused the observation in this study where there was no inhibition? Was this evaluated or only chanoclavine?

L422 – Change one Vmin to Vmax.

Fig. 1C – Is Control serotonin? Please indicate what control is in the figure legend.

Table 1 – Were all of these mutations performed in this study? What was the justification for not including more of them in Fig. 3?

Author Response

2# REVIEWER’S COMMENTS

# The work presented in this manuscript used the xenopus oocyte model as a tool to study the interaction of chanoclavine with the 5HT3A receptor. The work described appears to be done correctly with adequate replication. My main issue that must be addressed prior to acceptance is that the English needs to be improved throughout manuscript.

 Authors’ response:

First of all, thank you for expressing it as a well-done study and seeing it positively. The revised manuscript did proofread again by a professional English editing company. Also, we corrected specific comments which are your thorough review and salient observations.

All revised parts are highlighted in red. 

# Out of all of the ergot alkaloids to choose from to compare with chanoclavine, why was ergonovine selected? Why not another clavine alkaloid or an ergopeptine?

Authors’ response:

Ergot alkaloids act widely on various neuronal receptors. So we have been researching to reveal the mechanism by expressing various neuronal receptors in Xenopus oocytes in our previous papers. As a result, the paper published showing that Metergoline, identified as 5-HT1A agonist/5-HT2A antagonist, also works on Kv1.4. In the present study, it was confirmed that chnoclavine inhibits 5-ht3, so we expect a similar effect of ergonovine because ergonovine having a similar structure with chnoclavine. However it was not. In addition, the ergotpeptine group (ex Metergoline, Nicergoline), which has a large structural difference, did not work on 5-HT3A, and the other one chemical among Clavine groups has a modulating effect in the screening study. It is separately in the process of a paper, so it is difficult to reveal the name. Please understand this. In conclusion, the comparison object was selected as ergonovine because of the similar structure but showing completely different effects.

2# REVIEWER’S SPECIFIC COMMENTS

  1. L124 – Should the first figure referenced in the text be Fig. 1A? Perhaps reference Fig. 1A in the introduction where structural aspects are discussed such as L38 or L42.

Authors’ response:

We added the “Fig. 1A” [page 1, line 36]

  1. L314 – This statement made me wonder, what was different in the structure of ergonovine that caused the observation in this study where there was no inhibition? Was this evaluated or only chanoclavine?

Authors’ response:

Thank you for pointing it out. We found that one figure (figure 5) in the manuscript added mistakenly. We corrected Figure 5 to chanoclavine data, and the changes related to this were also revised in the manuscript. [page 9, line 255-257]

  1. L422 – Change one Vmin to Vmax.

Authors’ response:

Thank you for finding the missing parts when we checked.

We corrected Vmax [page 14, line 449]

  1. Fig. 1C – Is Control serotonin? Please indicate what control is in the figure legend.

Authors’ response:

We corrected from “Control” to “EG100uM” in Figure 1 files [page 4, line 131]

  1. Table 1 – Were all of these mutations performed in this study? What was the justification for not including more of them in Fig. 3?

Authors’ response:

Yes, we performed the with all mutants in table 1. Mutations not included in figure 3, are small or no active regulation. That's why we organized the rest mutants in table 1.

We hope this revised manuscript will meet your expectations. 

All revised parts are highlighted in red. 

Round 2

Reviewer 1 Report

Now the paper is suitable for pubblication 

Author Response

The manuscript by Eom and coworkers requires additional modifications before it can be accepted for publication. As pointed out by one of the reviewers, the manuscript needs to be improved in terms of English language and of technical language. With this respect, some points are highlighted below. However, additional improvements are needed. 

Authors’ response:

First of all, we appreciate your elaborate comments. We corrected all your comments. All revised parts are highlighted in red.

The title is too general. As the authors have studied two specific ergot alkaloids, the names of the compounds should appear in the title. 

Authors’ response: We corrected to “Differential regulation of human serotonin receptor type 3A by chanoclavine and ergonovine” [page 1, line 2-3]

Line 23-24: Docking modeling studies of chanoclavine confirmed that the 5-HT3A receptor was positioned between F130 and N138. - This statement makes no sense. Perhaps is chanoclavine positioned close F130 and N138 in 5 HT3A receptor binding site.

Authors’ response: We corrected them [page 1, line 23-24]

Line 27: Chanoclavine and ergonovine are similar in structure, but the opposite effects were seen, - Chanoclavine and ergonovine are similar in structure in a quite broad sense. It might be pretty obvious that the compounds have different effects.                       Authors’ response: We corrected to “have different effects” [page 1, line 27]

Line 28: suggesting that specific amino acid residues may specifically bind to the 5-HT3A receptor. This statement is not clear. 

Authors’ response: We corrected them to detail and for easily understand. [page 1, line 27-29]

Line 37: nitrogenous fungi Check this definition, please 

Authors’ response: We corrected to “fungi of the genus Claviceps.” [page 1, line 38-39]

Line 37: compounds induced by L-tryptophan Is the word induced appropriate?

Authors’ response: We corrected to “indole compounds produced from L-tryptophan” [page 1, line 38]

Line 41: can act as pharmacologically available receptor agonists or antagonists It is not clear the meaning of pharmacologically available

Authors’ response: We corrected to “indole compounds produced from L-tryptophan” [page 2, line 44-45]

Line 41: so these alkaloids are characterized as monoamine neurotransmitters Neurotransmitters are defined as the molecules used by the nervous system to transmit messages between neurons. Therefore, the term neurotransmitters are not appropriate for alkaloids. Authors’ response: We corrected to “so these alkaloids similarly act to the character of monoamine neurotransmitters” [page 1, line 43]

Line 47-48: The prevention of histamine and migraine that act as central sympathetic neuron activation and peripheral al-adrenergic blocking (6). This statement lacks the verb. The meaning is not clear. 

Authors’ response: We corrected to “The histamine that acts as central sympathetic neuron activation and peripheral α1-adrenergic blocking prevents migraine” [page 2, line 50-51]

Line 49-51: Bromocriptine promotes glutamate uptake by human glutamate transporter-1 and affects uterine contractions [7), which is effective for inhibiting prolactin release and treating Parkinson's disease (8-9). This statement should be rephrased as it is not clear the relationship between uterine contraction, inhibition of prolactin release, and Parkinson's disease. In addition, the effect on Parkinson's disease is related to potent D2 receptor agonism of bromocriptine. 

Authors’ response: We corrected to “Bromocriptine affects uterine contraction by promoting glutamate uptake in the human glutamate transporter -1 [7] and is related to Parkinson's disease through potent agonism on D2 receptor [8-9]” [page 2, line 51-53]

Line 54: delete "Therapeutic" 

Authors’ response: We deleted them [page 2, line 56]

Lines 58-62 

Serotonin receptors (5-HT, hydroxytryptamine) consist of a G-protein-coupled receptor and a ligand-gated ion channel type that exist in the central and peripheral nervous system and release or inhibit neurotransmitters by activation of their ligand, serotonin, which enhances neurotransmitter release including dopamine, cholecystokinin, and GABA (12]. The statement is too long and include various concepts. "Consists" should be replaced by "include". Specify which are GPCRs 5-HT receptors. 

Authors’ response: We corrected to “Serotonin (5-HT, hydroxytryptamine) receptors include seven families of 5-HT1, 5-HT2, 5-HT3, 5-HT4, 5-HT5, 5-HT6, and 5-HT7. Of these, only 5-HT3 is the only ligand-gated ion channel, and all other families are GPCRs. 5-HT receptors release or inhibit neurotransmitters (dopamine, cholecystokinin, and GABA) by activation of serotonin in the central and peripheral nervous system [12].” [page 2, line 60-65]

Line 67-69 The 5-HT3 receptor's main therapeutic target affects irritable bowel syndrome (IBS) in the vomiting center and gastrointestinal tract, and emesis caused by chemical triggers such as anticancer drugs. This statement is not written correctly. Please check.

Authors’ response: We corrected to “The main therapeutic target for the 5-HT3 receptor may be emesis induced in the vomiting center due to chemical triggers such as anticancer drugs” [page 2, line 70-71]

 Line 69-73 The activation of the 5-HT3 receptors in the peripheral nervous system performs a sensory function such as in the sympathetic and parasympathetic system and transports infor mation to the gastrointestinal tract and intestinal nervous system, which can regulate peristaltic movement and intestinal motility due to their role in causing emesis. 

The statement is too long and the meaning unclear. 

Authors’ response: We corrected to “The activation of the 5-HT3 receptors regulate peristaltic movement and intestinal motility by transporting sensory signals of the sympathetic and parasympathetic system to the gastrointestinal tract and the intestinal nervous system.” [page 2, line 72-74]

Line 74: perform emesis, recognition, anxiety, and depression functions Perhaps the line should be “is involved in emesis, recognition, anxiety, and depression.

Authors’ response: We corrected them to “is involved in emesis, recognition, anxiety, and depression.” [page 2, line 75-76]

Line 74-77; and are present at high levels in the brain stem. In particular, these receptors are present in areas such as the postrema and nucleus tractus solitarius related to vomiting reflex and in many brain areas such as the hippocampus, nucleus accumbens, and ventral tegmental area. Combine the information and split the text in two sentences.

Authors’ response: We corrected to “In particular, these receptors are present at high levels in the brain stem and expressed in many other brain regions such as the postrema hippocampus, nucleus accumbens, ventral tegmental area, and nucleus tractus solitarius. Among them, the hippocampus, nucleus accumbens, and ventral tegmental area relate to vomiting reflex.” [page 2, line 76-79] 

Line 77-79: Since serotonin mediates various reactions, multiple side effects can occur from insufficient or excessive serotonin in the human body Serotonin does no mediates reaction but physiological effects. Why side effects? Pathologies can arise from insufficient or excessive serotonin receptor stimulation in the human body? 

Authors’ response: We corrected them to “Since serotonin mediates various physiological effects, psychological disorders or serotonin syndrome can occur from insufficient or excessive serotonin in the human body” [page 2, line 80-81]

Line 79: specific subtype serotonin receptors Should be specific serotonin receptor subtypes 

Authors’ response: We corrected them to “specific serotonin receptor subtypes” [page 2, line 82]

Line 87-88: strong activity with rat Should be strong activity at rat 

Authors’ response: We corrected “with” to “at” [page 2, line 90]

Line 88: change results with data 

Authors’ response: We corrected them to “with data” [page 2, line 90]

Line 90-93 These are identified based on the structural principles obtained from functional studies such as molecular biology, second messenger, and ligand-binding analyses, demonstrating the applicability as a target for the treatment of vascular and nervous diseases and other disorders. These lines can be omitted. 

Authors’ response: We omitted them [page 2, line 92]

Line 105: the ligand-gated ion serotonin receptor Replace with 5-HT3 receptor

Authors’ response: We corrected them to “5-HT3 receptor” [page 3, line 104] 

Line 108: injecting human 5-HT3A receptor mRNA into Xenopus oocytes. 

Add a reference for this methodology. This is key for the whole study. 

Authors’ response: We added reference for injection methodology [page 3, line 107] 

Line 110-114 Although setron (a 5-HT3 antagonist) is widely used, the molecular binding and sup pression mechanisms are not known. In electrophysiology and in silico studies, docking simulation is essential to confirm the dominant residues that recognize setron binding and its ligand and help understand the inhibitory mechanism by comparing the setron and 5-HT binding structures A drug maned setron does not exist. The term setrons refers to 5-HT3 receptor antagonists (ondansetron, granisetron, etc.). This line must be corrected. In addition, the authors should check if docking studies on 5-HT3 receptor has been published so far. A comparison between their findings and the literature data would be welcomed. 

Authors’ response: We corrected them and added mention of comparison with existing literature (grainstron). Like this “We comparily analyzed the binding sites of granisetron (antagonist of 5-HT3A) and our chanoclavine through the existing literature” [page 3, line 110-112] 

Line 114: We investigated whether chanoclavine bound competitively to serotonin-binding sites and determined the location of the binding site using point mutations. Why ergonovine is not mentioned here?

Authors’ response: We added mention about ergonovine “and ergonovine” [page 3, line 109] 

Line 123: chanoclavine (EC) or ergonovine (EG), Once the abbreviations have been introduced, they should be used consistently from that poin of the manuscript on. 

Authors’ response: We deleted “(EC)” and “(EG)” [page 3, line 119] 

Line 168-169: This result indicates that chanoclavine interacted with the specific residue of the ligand-binding site. This should be read as "This result indicates that chanoclavine interacted with the same binding site of 5-HT" 

Authors’ response: We corrected them to “same binding site of 5-HT” [page 5, line 165] 

Line 171: Traces were induced by point mutations replaced by alanine in m5-HT3A Not clear the term traces here. Why m5-HT3A? Elsewhere in the manuscript one can read human receptor. 

Authors’ response: We corrected “m5-HT3A” to “h5-HT3A” [page 5, line 167] 

Lines 186: The average percentage curves of 5-HT3A mutant receptors expressing inward current fitted according to Hill's equation. This statement is not clear. What is Hill's equation for here? 

Authors’ response: We corrected to “The Inhibition percentage curve according to EC concentration of 5-HT3A mutant receptors.” [page 5, line 167] 

Line 242: what is covalent docking?

Authors’ response: We added the explanation of covalent docking. [page 9, line 237-238] Covalent docking is a method of predicting the interaction between a ligand and a receptor based on covalent bonds. Please see the literature for details:

‘Theory and Applications of Covalent Docking in Drug Discovery: Merits and Pitfalls’, Molecules 2015, 20, 1984-2000; doi:10.3390/molecules20021984

http://gohom.win/ManualHom/Schrodinger/Schrodinger_2015-2_docs/covalent_docking/covalent_docking_user_manual.pdf

Line 251: the 5-HT3A receptor forming with the extracellular pocket side. What is the extracellular pocket side? Please clarify.

Authors’ response: We corrected to “the binding pocket area on the extracellular side of the 5-HT3A receptor (Fig. 5A, B).” [page 9, line 250-251] 

Line 309-310: the 5-HT3A receptor plays an important role in regulating serotonin in the gastrointestinal tract and confirms its therapeutic effect. Check is 5-HT3 receptor regulates serotonin (release?) in the gastrointestinal tract. In addition, a receptor does not have a therapeutic effect per se. Instead, it can mediate a therapeutic effect. The sentence must be rephrased. 

Authors’ response: We corrected to “the binding pocket area on the extracellular side of the 5-HT3A receptor (Fig. 5A, B).” [page 9, line 250-251] 

Line 312: To demonstrate the mechanism and potential of ergot alkaloids as ligands, electrophysiology experiments and in silico docking simulation were performed - in silico docking simulation can suggest possible interactions in the binding site but does not demonstrate. The sentence must be rephrased. 

Authors’ response: We fully agree with your statement that docking simulations are computational predictions and only offer possibilities. So, we mainly focus on point-mutation and electrophysiology studies that can reveal actual interactions.

We corrected them to “We obtained candidates of binding sites by in silico docking simulation. To demonstrate the factual binding site of chanoclavine on 5-HT3A, Point mutation and electrophysiology experiments were performed.” [page 11, line 322-324] 

Line 313: In this study, it was thought that ergot alkaloids Change into "In this study, we have evaluated if ergot alkaloids" 

Authors’ response: We corrected them to “we have evaluated if ergot alkaloids” [page 11, line 324] 

Line 314 one of the neurotransmitter receptors with a similar structure. The structure of the receptor is not similar to the structure of alkaloids. Perhaps alkaloids have a structure that might resemble 5-HT. The sentence must be rephrased. 

Authors’ response: We corrected them to “because alkaloids have a structure that might resemble 5-HT” [page 11, line 325-326] 

Line 326: of 5-HT3A receptors, which have ligand-gated ion channels. Should be "of 5-HT3A receptor which is a ligand-gated ion channel. 

Authors’ response: We corrected them to “receptor, which is a ligand-gated ion channel.” [page 11, line 337-338] 

Line 335: ligand-binding site Delete ligand 

Authors’ response: We deleted “ligand” [page 11, line 346] 

Line 335: F130A and N138A, with different 5-HT3A sequences, were closely related to where the chanoclavine residues bound The meaning is not clear. What are 5-HT3A sequences?

We corrected them to “The amino acid sequences F130 and N138 of 5-HT3A receptor were closely related to where the chanoclavine bound residues” [page 11, line 346-348] 

Line 347: The site-directed mutagenesis method allowed us to identify the ligand-binding interaction sites with protein by shifting the channel amino acids to other properties. The 5-HT expression function remained unchanged and only the function of chanoclavine was lost so that F130 and N138 could be sites to which the chanoclavine residue bound. These lines are not clear. What is the ligand-binding interaction sites with protein? What are the other properties?

We corrected them to “The site-directed point mutagenesis method allowed us to identify the ligand-binding sites. We made the point-mutant 5-HT3A receptors by transforming all amino acids of candidates residue to alanine. Among them, two mutant-type (F130A and N138A) only lost the function of chanoclavine while remaing the function of serotonin.” [page 11-12, line 358-361] 

Remarks regarding docking studies.

1.Methods section does not contain all technical details needed to reproduce the results (e.g. Autodock settings).

We added setting information and Grid Box Information. [page 13, line 451-455] 

  1. The figures are not actually explaining the experimental data. The energy of ligand-protein contacts is explained only via the atom distances and no other details (such as predicted contact energies, docking scores, etc.) are provided. Moreover, it is difficult to correlate modeling data with activity, if the activity is equal to over 100 microM in terms of IC50). 

We added the data (binding energy, refRMS, pKi, etc.) [page 9, line 251-254]              Also, we added the mention that electrophysiology study is the main study and docking study is sub-study. [page 9, line 237-241], [page 11, line 322-324]

The results obtained in the manuscript should be compared with existing literature data on 5-HT3 receptor antagonist orientation in the binding pocket. In the literature, various examples of docking studies exist:

We agree with your mention, so we added comparison with chanoclavine and granisetron (existing literature data on 5-HT3 receptor antagonist) [page 9, line 263-270]

https://www.tandfonline.com/doi/abs/10.1081/rrs-120025568?journalCode=irst20 https://www.nature.com/articles/s41467-019-11142-8 https://www.jbc.org/article/S0021-9258(20)64047-9/fulltext http://fdjpkc.fudan.edu.cn/_upload/article/files/f5/3c/44f23e31420aaf4ee5836429281d/6be8f152-3d60-488a-b928-b7f9b898760d.pdf https://www.jneurosci.org/content/jneuro/24/41/9097.full.pdf https://europepmc.org/article/med/20409468 https://www.biorxiv.org/content/10.1101/2020.02.14.947937v1.full

We hope this revised manuscript will meet your expectations. 

All revised parts are highlighted in red. 

Reviewer 2 Report

Manuscript is much improved and is acceptable for publication.

Author Response

(The authors gave the same response as above.)
